# Dynamic Characteristics and Working Modes of Permanent Magnet Electrodynamic Suspension Vehicle System Based on Six Wheels of Annular Halbach Structure

Peng Lin, Zigang Deng *, Zhihao Ke, Wuyang Lei, Xuanbo Wang and Kehong Ren

State Key Laboratory of Traction Power, Southwest Jiaotong University, Chengdu 610031, China
* Correspondence: deng@swjtu.cn

**Abstract:** A novel type of suspension system for maglev vehicles using six permanent magnet electrodynamic wheels (EDW) and conductor plate has been designed. It has the advantages of high speed, environmental protection, and a low turning radius. Differing from existing maglev vehicles, this paper proposes a new maglev vehicle utilizing six EDWs to respectively provide driving force and levitation force. This structure can keep the levitation force at a large constant value and obtain enough driving force at low rotational speeds by adjusting the motor speed. First, the structure of the electrodynamic wheel is given. The accuracy and validity of the FEM results are verified by the experiments. Moreover, based on the finite element method (FEM), the optimal structure of the EDWs is obtained with the objective of maximum levitation force. Then, the simplified electromagnetic force model is obtained by using MATLAB Toolbox. Third, using a co-simulation of Simulink and Adams to design and build a 1:50 maglev vehicle model, this article studies the dynamic response characteristics of the maglev vehicle model from the perspective of dynamics and proposes a feedback control strategy by adjusting the rotational speed to control the maglev vehicle. This paper also proposes a method to realize the car's pivot steering to reduce the car's turning radius and help the drivers pass narrow road sections. This article verifies the feasibility of the maglev vehicle with six EDWs and is expected to provide a certain reference for the development of permanent magnet electrodynamic suspension vehicles.

**Keywords:** feedback control strategy; maglev vehicle; permanent magnet electrodynamic suspension; six permanent magnet wheels; turning radius

## 1. Introduction

Magnetic levitation is a mode of transportation whereby a vehicle is suspended above or below a track using the interaction of magnetic fields to achieve attraction or repulsion for both lift and thrust [1]. High-speed maglev ground transportation vehicles typically use high-temperature superconducting (HTS) flux pinning levitation, electromagnetic suspension (EMS), or electrodynamic suspension (EDS) [1–4]. The HTS system mainly consists of the high-temperature superconducting (HTS) levitation units (containing HTS bulks) attached to the vehicle and the permanent magnet guideway (PMG) on the ground [2,5]. It has many merits, such as the lack of wheel–rail adhesion contact, passive self-stability, potentially faster running speed, low energy consumption, and so on [2,6–8]. The EMS system makes use of a set of electromagnetic coils along the length of the vehicle which is attracted to a flat steel track and is actively controlled in order to maintain an air gap between the track and carriage. Furthermore, compared to EMS which uses attraction forces, the EDS system utilizes repulsive forces for levitation [3,9–11]. Either superconducting or neodymium iron boron (NdFeB) magnets on a carriage induce currents in the conductive coils or conductive plates along the track as the vehicle travels [12]. Because of its low cost, EDS is being studied as a substitute for existing high-speed maglev systems [13–15]. At

present, there are two main types of EDS technology: one is Superconducting Electrodynamic Suspension (SCEDS) and the other is Permanent Magnet Electrodynamic Suspension (PMEDS) [5,9]. Japan focused on low-temperature SCEDS technology in the 1970s [16]. It is used in the MLX maglev train. Through the use of the interactions between the superconducting magnets of the train and the "8"-shaped coils of the track, the suspension of the train is realized. As there is no wear or friction, the running speed of Maglev trains can reach more than 600 km/h [17]. PMEDS technology research has been developed in the 1980s with the advancement of NdFeB material production technology [9]. However, both PMEDS and SCEDS will create an unavoidable magnetic resistance. There are two ways to solve this problem. One is to mitigate this magnetic resistance, and the other is to convert the detrimental magnetic resistance into a useful propulsive force. Based on the second method, J. Bird et al. proposed a rotating annular Halbach structure electrodynamic wheel (EDW) at the end of the last century [18–20]. It is composed of EDW and a non-magnetic passive track that induces eddy current flowing. The Halbach magnet array can enhance the unilateral magnetic field and produce ideal magnetic field characteristics of the sinusoidal magnetic field waveform [21–25]. It can generate levitation force and magnetic resistance inside while EDW moves above a non-magnetic passive track [12]. By using EDWs as wheels to build a multi-EDW vehicle [3], it can convert the detrimental magnetic resistance into a useful driving force. Our group has studied the multi-EDW vehicle for several years. We have theoretically verified the feasibility of the multi-EDW vehicle and built a 1:50 PMEDS experimental prototype which has four EDWs [3]. However, the traditional four-wheel vehicle structure has a shortcoming: it can't realize the steering function. Because the traditional four-wheel structure changes the magnetic resistance by changing the rotational speed to form a torque, this method will cause the suspension force of the four EDWs to be inconsistent, resulting in the change of the suspension air gap, which will affect the smoothness of the vehicle operation and lead to steering failure. To solve the steering problem of maglev vehicles, in this paper, we design a new concept maglev vehicle structure based on the radial Halbach permanent magnet array. Figure 1 shows its concept diagram. Although the use of six wheels will increase the overall cost of the vehicle, considering that this structure can help the car achieve steering, these increased costs are worthwhile, and the increase of guided wheels will not have any impact on the driving safety of the car. At the same time, the non-magnetic passive track structure can be more easily integrated into the existing traffic infrastructure. The construction cost will not increase too much. In addition, a novel control strategy is proposed in this paper to realize straight-line motion and pivot steering.

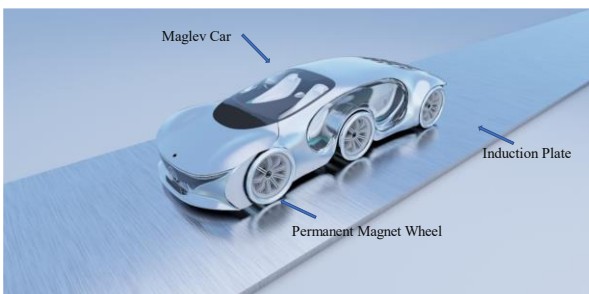

**Figure 1.** Concept diagram of the permanent magnet electrodynamic suspension vehicle with six wheels.

The rest of this paper is organized as follows. Initially, the structure and principle of the operation of the EDW are illustrated in Section 2. Subsequently, the accuracy analysis of constructed FEM will be completed compared with the experiment in Section 4. Then, the related parameters of the selected optimal structure of the EDW will also be clarified in Section 5. Eventually, a novel control strategy is verified by the co-simulation of Simulink and Adams Platform, which could facilitate the vehicle system to realize a stable straight-

line motion and pivot steering. With the development of society, there will be more and more vehicles in the city, but the speed of road construction cannot keep up with the growth rate of vehicles. Especially in some cities' centers, the roads are relatively narrow and unable to be rebuilt. Therefore, this new maglev can relieve the traffic pressure by reducing the car's turning radius.

## 2. Structural Design

Figure 2 shows the relationship of levitation force, thrust force, and rotational speed. The driving force and the levitation force have a strong coupling relationship [3,9,10,26–28]. With the increase of rotational speed of the wheel, the levitation force generated by the eddy current field increases gradually, but the growth rate of the levitation force slows down with the increase of rotational speed [9,29,30]. The thrust force will quickly reach the peak as the speed rotational increases [19]. When the rotational speed reaches an inflection point, the thrust force gradually decreases and tends to be stable, while the levitation force continues to increase; therefore, it is difficult to provide enough driving force and levitation force at the same time with one EDW [15,31]. If the rotation speed decreases, the levitation force will decrease and the air gap also needs to decrease to maintain enough levitation force.

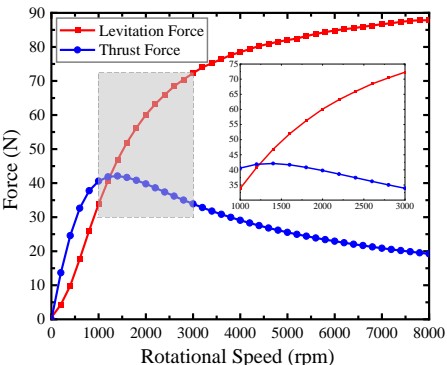

**Figure 2.** Change of the levitation force and driving force with the rotational speed of the EDW.

So if the traditional four-wheel vehicle structure is used to drive the maglev vehicle by adjusting the relative rotational speed of the front and rear wheels [3], there will be two more disadvantages: First, when the maglev vehicle is accelerated by adjusting the rotational speed, it is necessary for the traditional four-wheel vehicle structure to decrease the rotational speed of the front EDWs in order to obtain enough driving force [3]. However, this will decrease the levitation force and the suspension gap of the front wheels. If the suspension gap decreases, the driving safety of the vehicle cannot be ensured. Second, the relative rotational speed of the front and rear wheels is not equal when the maglev vehicle is accelerated and decelerated, therefore the suspension force generated by the front and rear wheels are different and a torque is formed, which makes the maglev vehicle have a trend of pitch movement and it becomes difficult to run smoothly. Third, although a torque can be formed by decreasing the rotational speed of the EDWs on the diagonal, the suspension air gaps of the four EDWs are different, which will affect the smoothness of the vehicle operation and lead to steering failure.

Based on the principle of EDS, this article subverts the current situation of traditional EDS technology, and a PMEDS structure based on the annular Halbach structure is proposed. Figure 3 shows its structure diagram. The innovation of this structure is to separate the EDWs that provide driving force and levitation force. The six wheels of the car will be divided into suspended wheels and guided wheels. The EDWs in the middle of the vehicle will provide the driving force for acceleration, deceleration and steering. The EDWs of the front and rear will only provide levitation force for the vehicle. This structure takes advantage of the fact that the magnetic resistance is larger at low speeds and smaller at

high speeds. The high-speed magnetic wheels are used to provide levitation and reduce the energy loss caused by magnetic resistance. The less energy loss meets the standards of sustainable transportation. The low-speed magnetic wheels providing the driving force convert the inherent magnetic resistance into driving force, so as to realize the non-contact running of the suspension system. Moreover, the driving energy of the car comes from the rotating motor, which does not use fossil energy. It also meets the requirements of environmental protection, so this maglev vehicle is suitable in the context of sustainable transportation.

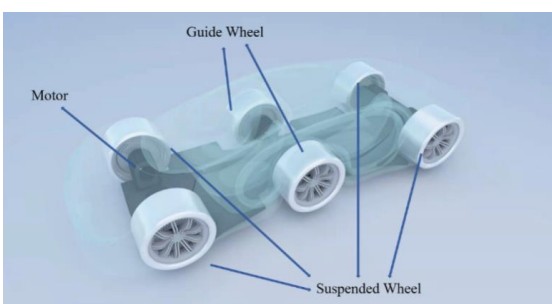

**Figure 3.** Structure diagram of the permanent magnet electrodynamic suspension vehicle.

At the same time, this paper proposes a feedback control strategy, which can make the control system form a closed loop. This paper will use a 1:50 PMEDS car model. The aggregate mass of the PMEDS vehicle is 14.52 kg. The suspension drive section of this vehicle is composed of six EDWs and six motors. The permanent magnet wheel adopts the annular Halbach structure.

## 3. Principle of the Operation

When the EDW is stationary, the relative speed of the magnetic wheel and the conductor plate is the self-rotational speed of the magnetic wheel, but when the EDW has speed in the horizontal direction, the relative speed of the magnetic wheel and the induction plate is simultaneously determined by the self-rotational speed of the magnetic wheel and translational motion of the EDW. This problem is illustrated in Figure 4 [32–34].

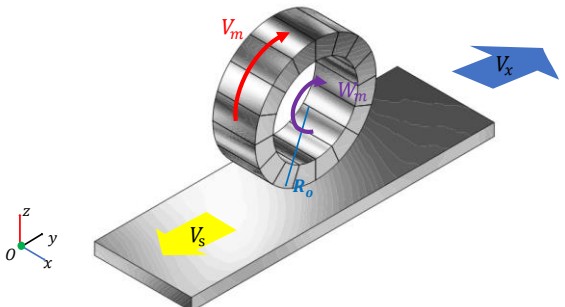

**Figure 4.** Schematic diagram of the EDW with a conductor plate in the horizontal direction.

The EDW has a translational speed of $V_x$. Moreover, the angular velocity of the EDW rotation is $W_m$, the outer radius is $R_o$, and the relative speed between the wheel and the plate is $V_s$, which satisfies the following relationship [19,35]:

$$V_s = V_m - V_x \tag{1}$$

$$V_m = W_m R_o \tag{2}$$

When $V_s$ is greater than or less than 0, the EDW will be in the state of acceleration or deceleration braking mode. Furthermore, when $V_s = 0$, the EDW will remain uniform motion state.

Figure 5 shows the operating principle of the vehicle model. The front EDWs turn clockwise at the rotational speed of $w_1$ and the direction of magnetic resistance $F_{d_1}$ and $F_{d_3}$ are backward, and the rear EDWs turn counterclockwise at the rotational speed of $w_2$ and the direction of magnetic resistance $F_{d_2}$ and $F_{d_4}$ are forward. The suspension function is achieved by synchronizing the rotational speed of the front and rear EDWs, turning in opposite directions, generating driving forces of the same size and opposite directions. The levitation forces $F_{L_1}$, $F_{L_2}$, $F_{L_3}$, and $F_{L_4}$ generated by the front and rear EDWs overcome the weight of the entire device to achieve pivot suspension. Then tangential forces $F_{d_5}$ and $F_{d_6}$ are generated by the rotation of the middle wheel to realize the driving function and steering function. When the car is driving at a constant speed or is static levitation, the relative rotational speed $w_3$ of the middle wheels is 0.

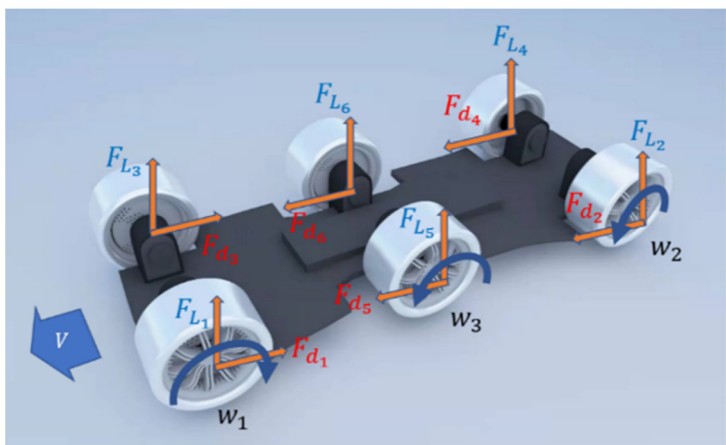

**Figure 5.** Principle of the operation of the vehicle model.

## 4. EDW Design, Verification, and Optimization

The main core of the electromagnetic vehicle model is an induction eddy current model composed of a moving permanent magnet and an induction plate. Presently, the most popular method for calculating a Halbach array is to use finite element software for simulation and design [36]. This paper uses finite element software Maxwell for numerical calculations. Maxwell transient solver is selected to better reflect the dynamic characteristics of the suspension system. A simplified three-dimensional model of the EDW was established, as shown in Figure 6. The specific parameters of the permanent magnet EDW used are shown in Table 1.

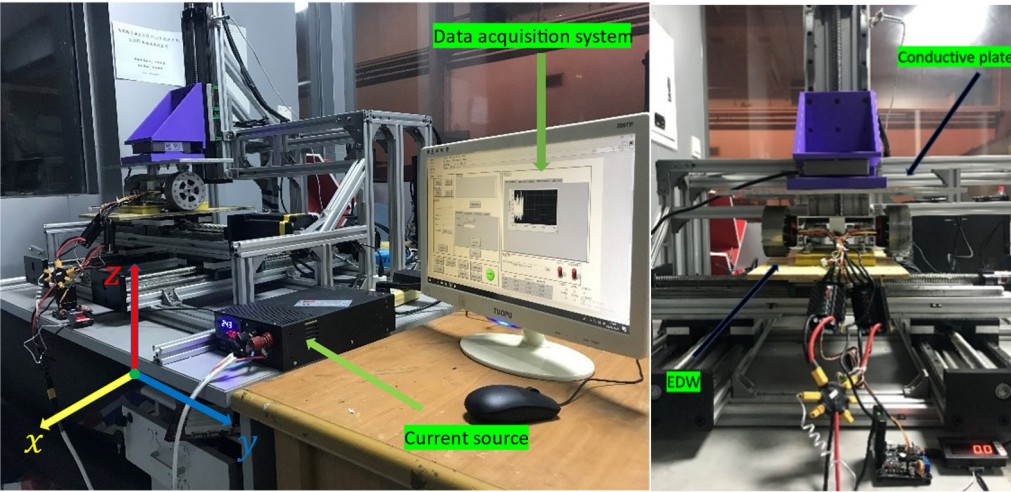

**Figure 6.** Image of SCML-05 test platform for EDW.

**Table 1.** Parameters of the permanent magnet EDW.

| Parameter | Value |
|---|---|
| Pole pairs, P | 4 |
| Remanence of permanent, $B_r$/T | 1.42 |
| Inner radius of EDW, $R_i$/mm | 35 |
| Outer radius of EDW, $R_o$/mm | 50 |
| Width of EDW, $W_w$/mm | 35 |
| Width of Conductor Plate, $W_p$/mm | 70 |
| Thickness of Conductor Plate, $T_p$/mm | 10 |
| Resistivity of Conductor Plate, $\rho$/$\Omega$·m | $2.826 \times 10^{-8}$ |
| Velocity of motor, V/rpm | 0–6000 |
| Air gap, G/mm | 3–15 |
| Magnetization angle, A/° | 90 |

To determine the accuracy of FEM calculations, this paper uses the SCML-05 test system for verification. It was developed by our group and is used for 3D axial forces tests. The SCML-05 is shown in Figure 6. It is mainly composed of four stepper motors and a data acquisition system. In the test within this paper, the test system is used to measure the vertical levitation force and the horizontal thrust force of the EDW. The data acquisition system uses the software designed by LabVIEW. It can display, collect, and export force data and distance data in three directions.

The following briefly covers some important steps for experimenting with the EDW on the SCML-05 test platform.

1.  Assembly:
    Firstly, fix the EDWs with a fixture, and assemble them to the horizontal platform at the bottom of SCML-05. Ensure that the conductor plate is fixed horizontally so that the measuring path of the force is parallel to the cross-section of the conductor plate [9].
2.  Positioning:
    Start the SCML-05 test system, adjust the conductor plate to the initial levitation height of 10 mm in the Z-direction. Then confirm that the EDW structure is horizontally placed under the conductor plate, and the EDWs and conductor plate should be parallel.
3.  Test:
    After setting up the test system, the test is carried out at the different selected speed point. The speeds, levitation forces and driving forces should be recorded when the sensor reading is stable.

In the experiment, the average value of multiple measurements should be taken to reduce the experimental error. Due to the eddy current loss in the experiment, the temperature of the conductor plate will inevitably continue to rise. In severe cases, the resistivity of the conductor plate may increase, which will affect the experimental results. Therefore, it is very necessary and important to shorten the collection time of data as much as possible in the experiment. When the air gap is 10 mm, the levitation force and thrust force at different rotational speeds are simulated. As shown in Figure 7, the experimental data at different speeds are compared with the 3D FEM results. It is found that the simulation results of the levitation forces are larger than the experiment data, with an average error of about 2.79%. The simulation results of the driving forces are smaller than the experiment data, with an average error of about 11.05%.

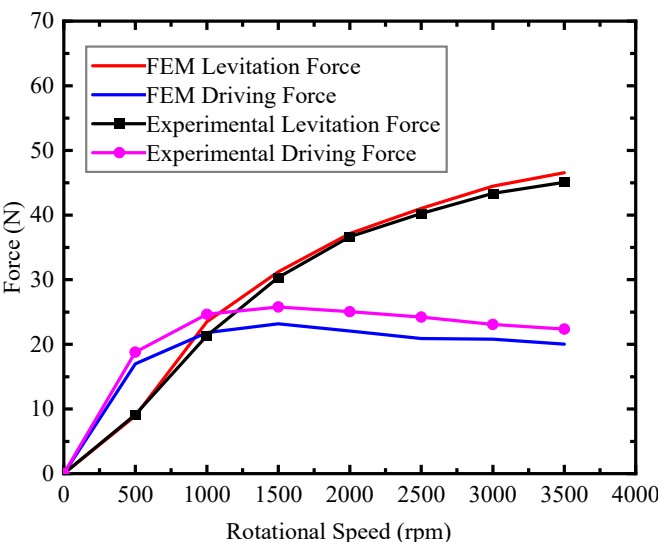

**Figure 7.** Comparison of experimental data and simulation results of levitation and driving forces of the EDW with different rotational speeds when the air gap is 10 mm.

The reason for the difference between the experimental data and the 3D FEM results is that the resistivity of the aluminum used in the FEM is $2.826 \times 10^{-8}$ $\Omega$·m, but the conductor plate used in the experiment is not pure aluminum, which has a larger resistivity. This will make the levitation force smaller and the driving force larger [32,37]. However, the variation trends of the levitation force and driving force obtained by the experiment are consistent with the simulation, indicating that when the air gap is constant, the EDW can provide stable suspension force and driving force to achieve suspension and driving functions.

The experiment results have proved the reliability of the FEM. However, the four EDWs with four pole pairs and 90-degree magnetization angle can just provide about 144 N levitation force for the maglev vehicle with four EDWs when the air gap is 10 mm. The aggregate mass of the PMEDS vehicle is about 14.52 kg. Therefore, they cannot provide enough levitation force when the air gap is larger. When the road is irregularity, the vehicle body will vibrate vertically. If the air gap is too small, the EDWs of maglev vehicle will collide with the conductor plate. In order to ensure the driving safety of the vehicle, the air gap of the maglev vehicle must be increased. We hope that the maglev car can reach a suspension height close to 14 mm. It is necessary to improve the levitation force of the EDWs. The levitation force of a circular Halbach array is affected by many factors. The size, material, magnetization angle of the magnet, and the number of pole pairs are factors that affect the levitation force of the Halbach array [38]. However, in order to improve the suspension height, we need to improve the suspension force without changing the wheel weight. Increasing the size of magnets can improve the levitation force, but it will increase the weight of the wheel and increase the cost; selecting better magnet materials can also increase the levitation force, but this kind of magnet is expensive and difficult to produce. According to previous research, there are two methods to achieve this goal. One is to change the number of pole pairs in one EDW [11,39], the other is to change the magnetization angle of the magnet. In this article, we choose to optimize the magnetization angle and pole pairs at the same time to find a magnetic wheel that satisfies the requirements.

The results of the relationship between the levitation force and the pole pairs of the EDW are presented in Figure 8. The levitation force decreases as the number of pole pairs increases. The EDWs with two pole pairs can provide the largest levitation force, but it is not stable, because its frequency of magnetic field changes is too slow. It is better to choose the three pole pairs EDWs.

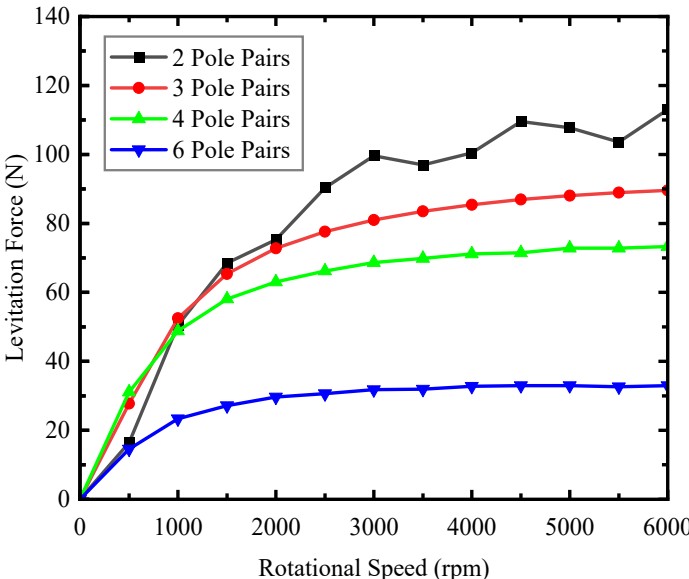

**Figure 8.** Relationship between the levitation force and the pole pair of the EDW.

The results of the relationship between the levitation force and the magnetization angle of the EDW are presented in Figure 9. The levitation force increases as the magnetization angle decreases. This is because the magnetic field becomes closer and closer to the ideal Halbach array with the decrease of the magnetization angle. However, magnets with a magnetization angle of less than 30 degrees are difficult to process, so a magnetization angle of 30 degrees was selected in the end. Due to the experimental conditions, there is no way to perform verification experiments for the magnetic wheel with a 30-degree magnetization angle and three pole pairs. The previous experiments have verified the correctness of the finite element results, so this paper will use EDW with three pole pairs and 30 degrees. The specific parameters of the permanent magnet EDW finally used are shown in Table 2. The FEM results are presented in Figures 10 and 11.

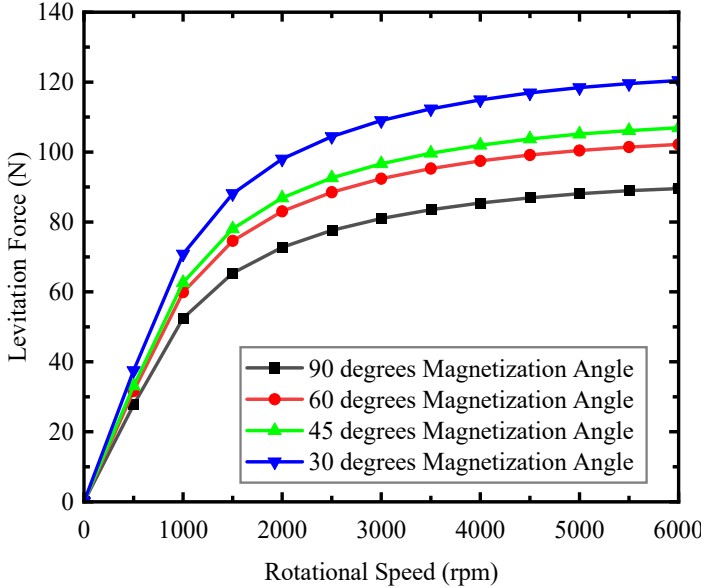

**Figure 9.** Relationship between the levitation force and the magnetization angle of the EDW.

**Table 2.** Specific Parameters of The Permanent Magnet EDW.

| Parameter | Value |
|---|---|
| Pole pairs, P | 3 |
| Remanence of permanent, $B_r$/T | 1.42 |
| Inner radius of EDW, $R_i$/mm | 35 |
| Outer radius of EDW, $R_o$/mm | 50 |
| Width of EDW, $W_w$/mm | 35 |
| Width of Conductor Plate, $W_p$/mm | 70 |
| Thickness of Conductor Plate, $T_p$/mm | 10 |
| Resistivity of Conductor Plate, $\rho$/Ω·m | $2.826 \times 10^{-8}$ |
| Velocity of motor, V/rpm | 0–6000 |
| Air gap, G/mm | 3–15 |
| Magnetization angle, A/° | 30 |

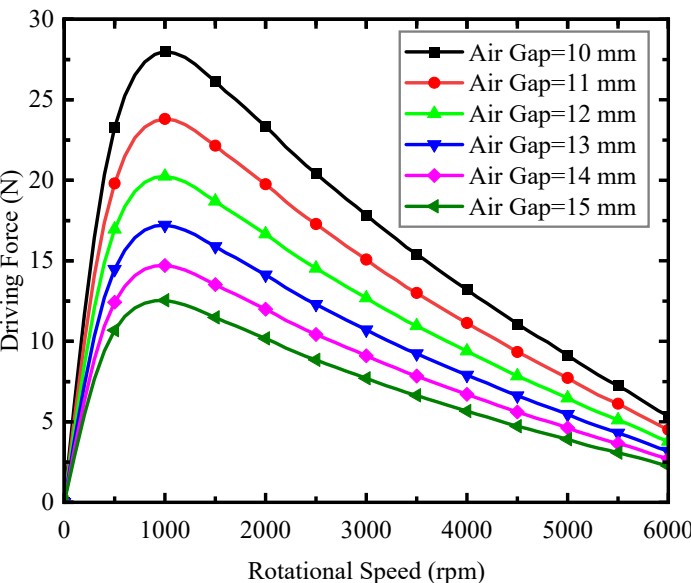

**Figure 10.** Relationship of the driving force with rotational speed of the EDW at different air gaps.

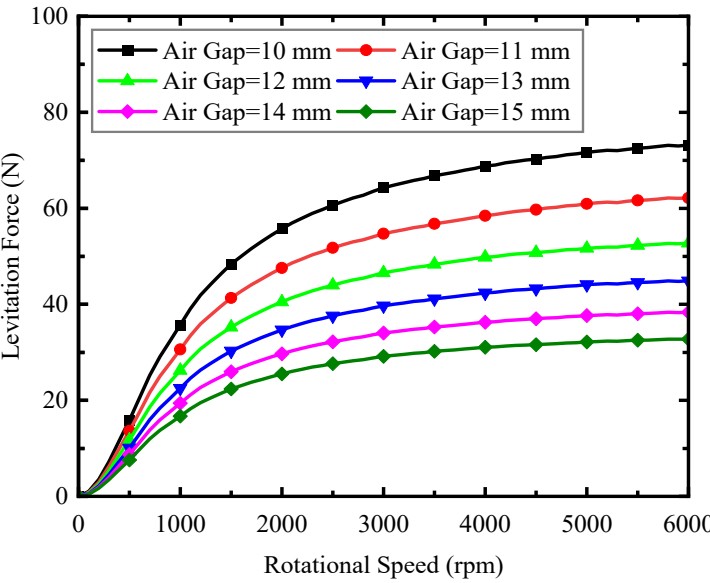

**Figure 11.** Relationship of the levitation force with rotational speed of the EDW at different air gaps.

Figure 10 shows that the critical speed of the driving force is about 1000 rpm. Figure 11 shows when the air gap is 14 mm and the rotational speed is 4000 rpm, one EDW can provide about 36 N levitation force. The aggregate mass of the PMEDS vehicle is about 14.52 kg. Therefore, a maglev car with four suspended wheels can achieve a suspension height of about 14 mm.

## 5. Simplified Electromagnetic Force Model

To provide electromagnetic force input for dynamic simulation and accurately analyze the dynamic response of the PMEDS system, it is necessary to establish a simplified electromagnetic force model. MATLAB Toolbox has the characteristic of convenient solution and high accuracy [3]. Based on the force data obtained by FEM, this paper chooses the polynomial fitting method to obtain the expression of the driving force and levitation force that will be used in the dynamic simulation. With the parameters of the suspension system fixed, the levitation force and driving force are a binary function of the rotational speed of the magnetic wheel and the suspension air gap [3]. At the same time, to better fit the data, this paper also chooses Center and Scale when using the polynomial fitting method. The equations used to Center and Scale can be obtained as follows:

$$x' = \frac{x - mean(x)}{std(x)} \tag{3}$$

$$y' = \frac{x - mean(y)}{std(y)} \tag{4}$$

The parameters in the equations used to Center and Scale are summarized in Tables 3 and 4.

**Table 3.** Parameters in the Equations Used to Center and Scale of Levitation Force $F_L$.

| Parameter | Value |
|---|---|
| Rotational Speed of the EDW, $x'$/rpm | 0–6000 |
| Suspension Air Gap, $y'$/mm | 3–15 |
| Rotational Speed of the EDW before Center and Scale, $x'$/rpm | 0–6000 |
| Suspension Air Gap before Center and Scale, $y'$/mm | 3–15 |
| Average of the $x$, $mean(x)$/rpm | 3000 |
| Average of the $y$, $mean(y)$/mm | 9 |
| Standard Deviation of the $x$, $std(x)$/rpm | 1761.793 |
| Standard Deviation of the $y$, $std(y)$/mm | 3.744 |

**Table 4.** Specific Parameters of the Permanent Magnet EDW.

| Parameter | Value |
|---|---|
| Rotational Speed of the EDW, $x'$/rpm | 0–6000 |
| Suspension Air Gap, $y'$/mm | 3–15 |
| Rotational Speed of the EDW before Center and Scale, $x'$/rpm | 0–6000 |
| Suspension Air Gap before Center and Scale, $y'$/mm | 3–15 |
| Average of the $x$, $mean(x)$/rpm | 3600 |
| Average of the $y$, $mean(y)$/mm | 9 |
| Standard Deviation of the $x$, $std(x)$/rpm | 2114.15 |
| Standard Deviation of the $y$, $std(y)$/mm | 3.744 |

The mathematical models of the levitation force $F_L$ and driving force $F_D$ obtained by polynomial fitting method are:

$$\begin{aligned} F_L = \ & L_{00} + L_{10}x + L_{01}y + L_{20}x^2 + L_{11}xy + L_{02}y^2 + L_{30}x^3 + L_{21}x^2\,y + \\ & L_{12}xy^2 + L_{03}y^3 + L_{40}x^4 + L_{31}x^3y + L_{22}x^2y^2 + L_{13}x\,y^3 + L_{04}y^4 \end{aligned} \tag{5}$$

$$\begin{aligned}
F_D = \quad & D_{00} + D_{10}x + D_{01}y + D_{20}x^2 + D_{11}xy + D_{02}y^2 + D_{30}x^3 + D_{21}x^2y + \\
& D_{12}xy^2 + D_{03}y^3 + D_{40}x^4 + D_{31}x^3y + D_{22}x^2y^2 + D_{13}xy^3 + D_{50}x^5 + \\
& D_{41}x^4y + D_{32}x^3y^2 + D_{23}x^2y^3
\end{aligned} \tag{6}$$

The parameters in the mathematical models of the levitation force $F_L$ and driving force $F_D$ obtained by the polynomial fitting method are summarized in Tables 5 and 6.

**Table 5.** Parameters in the Mathematical Models of the Levitation Force $F_L$ Obtained by Polynomial Fitting Method.

| Parameter | Value |
|---|---|
| $L_{00}$ | 75.95 |
| $L_{10}$ | 10.55 |
| $L_{01}$ | $-48.69$ |
| $L_{20}$ | $-8.31$ |
| $L_{11}$ | $-8.171$ |
| $L_{02}$ | 16.41 |
| $L_{30}$ | 6.022 |
| $L_{21}$ | 8.237 |
| $L_{12}$ | 5.66 |
| $L_{03}$ | $-3.351$ |
| $L_{40}$ | $-1.369$ |
| $L_{31}$ | $-3.183$ |
| $L_{22}$ | $-2.6$ |
| $L_{13}$ | $-1.437$ |
| $L_{04}$ | 0.465 |

**Table 6.** Parameters in the Mathematical Models of the Driving Force $F_D$ Obtained by Polynomial Fitting Method.

| Parameter | Value |
|---|---|
| $D_{00}$ | 19.6 |
| $D_{10}$ | $-8.085$ |
| $D_{01}$ | $-13.26$ |
| $D_{20}$ | 8.853 |
| $D_{11}$ | 10.28 |
| $D_{02}$ | 6.299 |
| $D_{30}$ | $-5.701$ |
| $D_{21}$ | $-4.528$ |
| $D_{12}$ | $-3.497$ |
| $D_{03}$ | $-1.572$ |
| $D_{40}$ | $-5.037$ |
| $D_{31}$ | $-3.709$ |
| $D_{22}$ | $-1.245$ |
| $D_{13}$ | 0.2779 |
| $D_{50}$ | $-3.203$ |
| $D_{41}$ | 2.884 |
| $D_{32}$ | 1.236 |
| $D_{23}$ | 0.3212 |

As shown in Figures 12 and 13, the red curves obtained by the finite element software are highly coincident with the blue curves obtained by the polynomial fitting method. This proves the reliability of mathematical models of the levitation force $F_L$ and driving force $F_D$.

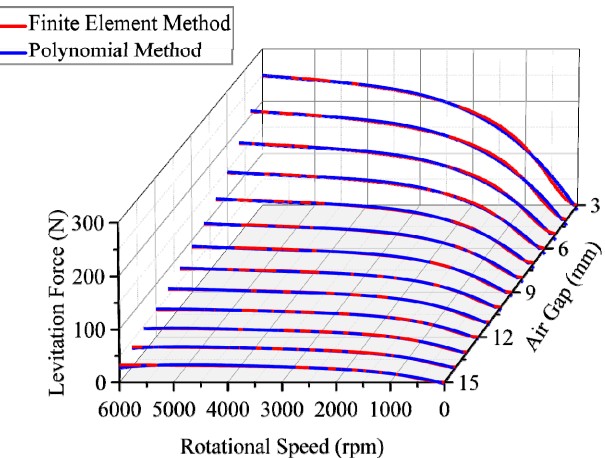

**Figure 12.** Comparison of polynomial method results and FEM results of levitation force with different rotational speeds and different air gaps.

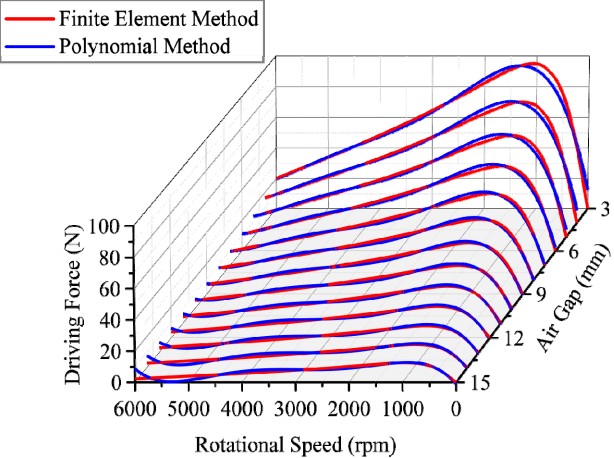

**Figure 13.** Comparison of polynomial method results and FEM results of driving force with different rotational speeds and different air gaps.

## 6. Establishment of The Simulation Platform

To find a suitable control strategy to control the movement of the maglev car, it is necessary to find a way to directly observe the changes in the suspension air gap, levitation force, driving force, and translational speed of the car in real time and adjust the movement state of the car.

Adams is a software that was developed based on the Lagrangian equation method of multi-body system dynamics and integrates system modeling, solving, and visualization technologies. Users can use Adams to study the dynamics of moving parts, and how loads and forces are distributed throughout mechanical systems. It has been widely used in the automotive industry. At the same time, its open program structure and multiple external interfaces can provide a secondary development research platform to analyze special types.

Simulink is a MATLAB-based graphical programming environment for modeling, simulating, and analyzing multidomain dynamical systems. Its primary interface is a graphical block diagramming tool and a customizable set of block libraries. It is widely used in automatic control and digital signal processing for multidomain simulation and model-based design [3].

This paper uses the co-simulation of Simulink and Adams, which are based on the polynomial fitting of the electromagnetic force, to find a suitable feedback control strategy to control the movement of the maglev car. Figure 14 shows the principle of establishment of the co-simulation of the Simulink and Adams platform.

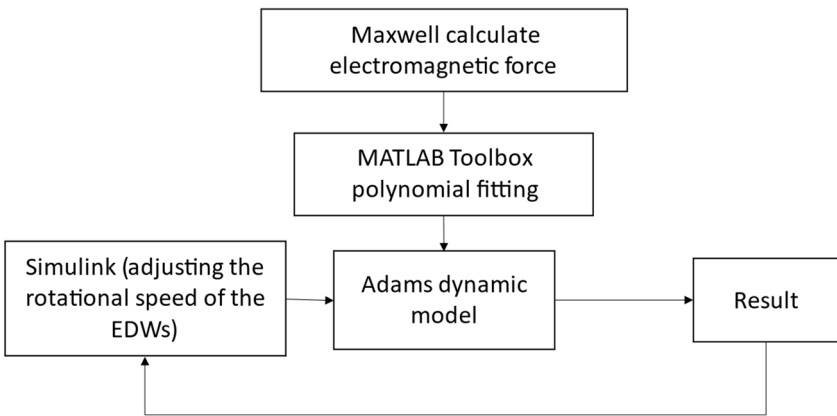

**Figure 14.** Principle of the establishment of the co-simulation of Simulink and Adams platform.

## 7. Control Strategy

To control the maglev car, this paper proposes a feedback control strategy by adjusting the rotational speed to control the maglev vehicle. It is necessary to use the output of the system and make adjustments in time to form a feedback control. This article selects the relative rotational speed of the EDWs. By converting the translational speed of the maglev vehicle into the rotational speed compensation of the EDW, the relative rotational speed of the front and rear EDWs will always be maintained at 4000 rpm and the relative rotational speed of the middle EDWs will maintain at 0 rpm. When the maglev vehicle needs to accelerate, decelerate, and turn, the speed of middle EDWs will temporarily not be equal to 0 rpm. However, after adjusting the rotational speed of the EDWs through Simulink, the maglev vehicle will enter a stable state again. This control strategy is simpler and easier to implement. The control flow graph is shown in Figure 15.

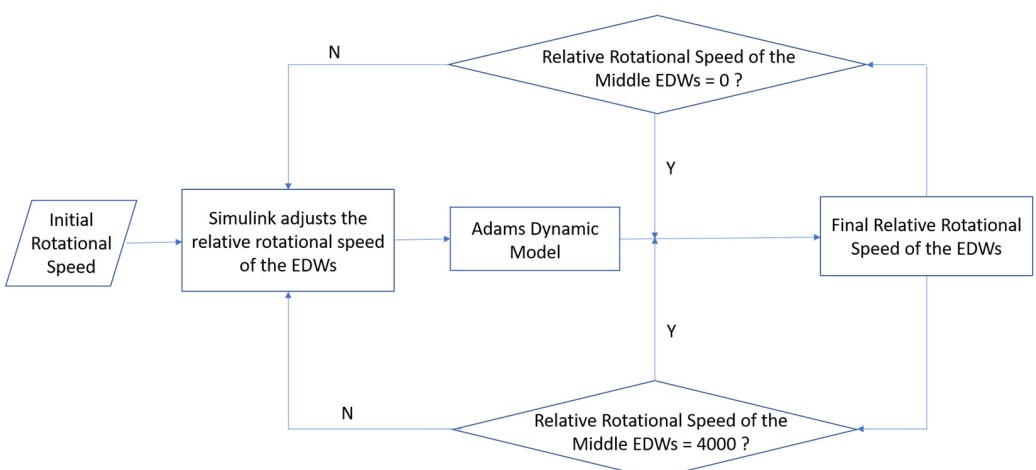

**Figure 15.** Control Flow Graph of Co-simulation.

### 7.1. Static Suspension

To achieve static suspension, the rotational speed of the front and rear EDWs are the same and the direction is reversed, when the maglev vehicle is at static suspension. By increasing the motor speed gradually from 0 to 4000 rpm, the PMEDS vehicle achieves suspension from stationary on the ground. During this period, the speed of the middle EDWs relative to the induction plate remains at 0 rpm. Figure 16 shows the diagram of the PMEDS car model. The rotational speed of the EDWs and suspension height of the maglev car and the levitation force of all the EDWs during the static suspension process are shown in Figures 17 and 18, respectively.

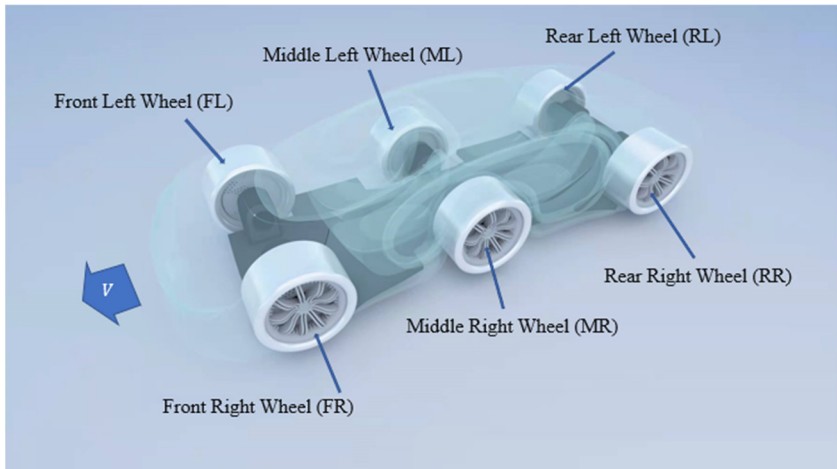

**Figure 16.** The schematic diagram of the PMEDS car model.

From Figure 17, it can be seen that the levitation force of the EDW gradually increases with the increase of speed before the suspension. After the EDW speed reaches about 280 rpm (after 2.6 s), the levitation force of the EDWs can balance the deadweight of the whole system, and the maglev vehicle achieves suspension. The aggregate mass of the PMEDS vehicle system is about 14.52 kg. As the EDWs' speed continues to increase, the levitation force which balances with gravity will maintain at about 145 N. As a result, the suspension air gap will increase to balance the levitation force. Since there is no resistance in the dynamic model, the damping of PMEDS is almost zero [3]. The levitation force and the suspension air gap will keep fluctuating and cannot be stabilized, and the suspension air gap will remain at about 14 mm.

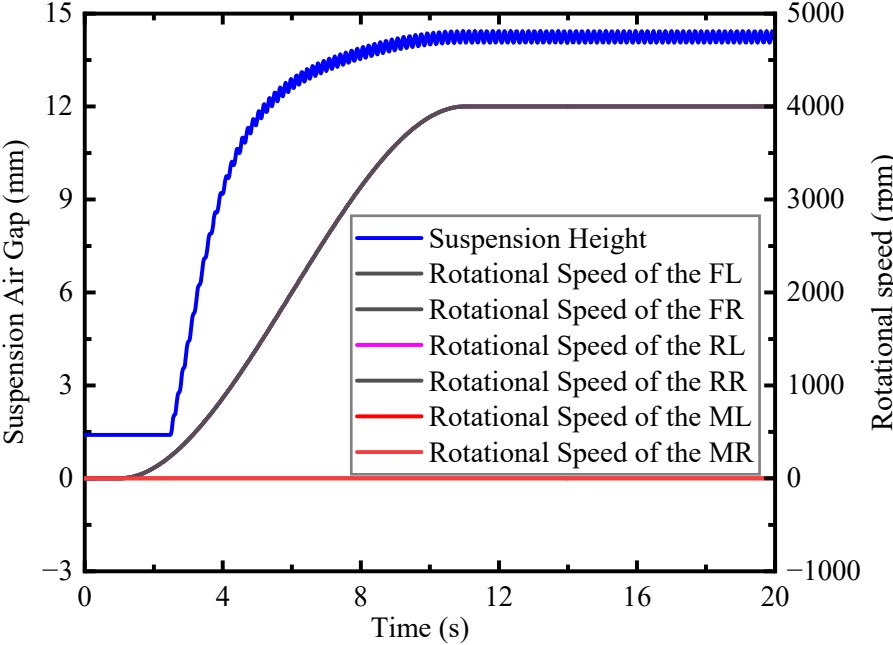

**Figure 17.** Rotational speed of the EDWs and suspension height of the maglev car during static suspension processes.

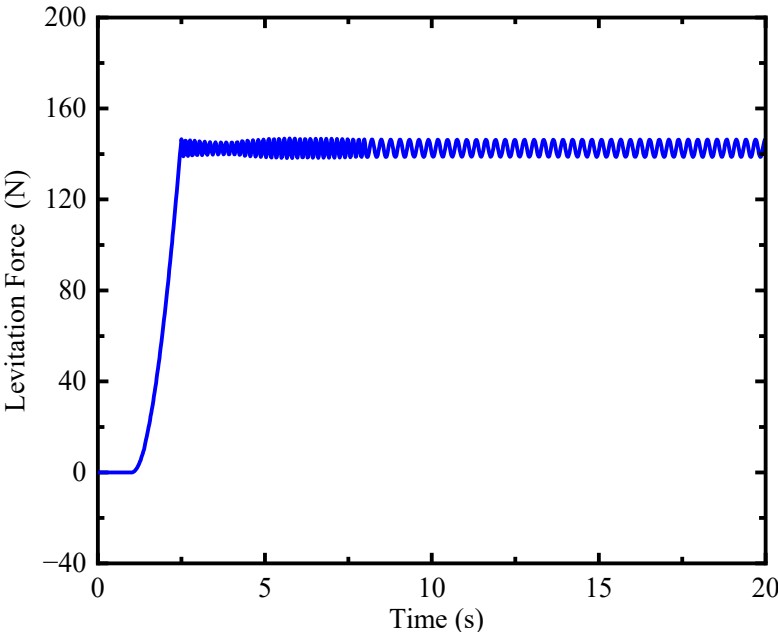

**Figure 18.** Levitation force of all the EDWs during static suspension processes.

### 7.2. Acceleration and Uniform Speed

The method of acceleration is to keep the relative rotational speed of the front and rear wheels relative to the ground at 4000 rpm, so that the driving force of the front EDWs is equal to the driving force of the rear EDWs. At the same time, the rotational speed of the middle EDWs is increased to generate the driving force and drive the maglev vehicle to accelerate.

The relative rotational speed and rotational speed regulation process of the middle EDWs during acceleration and uniform speed is shown in Figure 19. The relative rotational speed and rotational speed of the front and rear EDWs and the translational speed of the maglev vehicle are shown in Figures 20 and 21, respectively. From 1 s to 11 s, the rotational speed of the front and rear EDWs increases from 0 to 4000 rpm, and the maglev vehicle goes from a static state to a suspended state. From 16 s to 17 s, the rotational speed of the middle EDWs increases by 500 rpm, and the maglev vehicle begins to accelerate. But the rotational speed of the middle EDWs cannot reach 500 rpm due to the existence of the vehicle's translational speed and feedback regulation during this period. The rotational speed of the front wheels increases and the rotational speed of the rear wheels decreases to keep the relative rotational speed of the front and rear EDWs relative to the conduction plate remain 4000 rpm.

From 17 s to 20 s, the relative rotational speed of the middle EDWs continues to decrease until it reduces to 0 and the driving force also becomes 0 at 20 s, and the speed of the maglev vehicle stabilizes at 1.31 m/s and begins to drive at a constant speed. Because this article uses a 1:50 model, the final speed of the maglev vehicle is not high. If the rotational speed of the middle wheel is further increased and using a full-scale model, the final speed of the maglev vehicle can be increased.

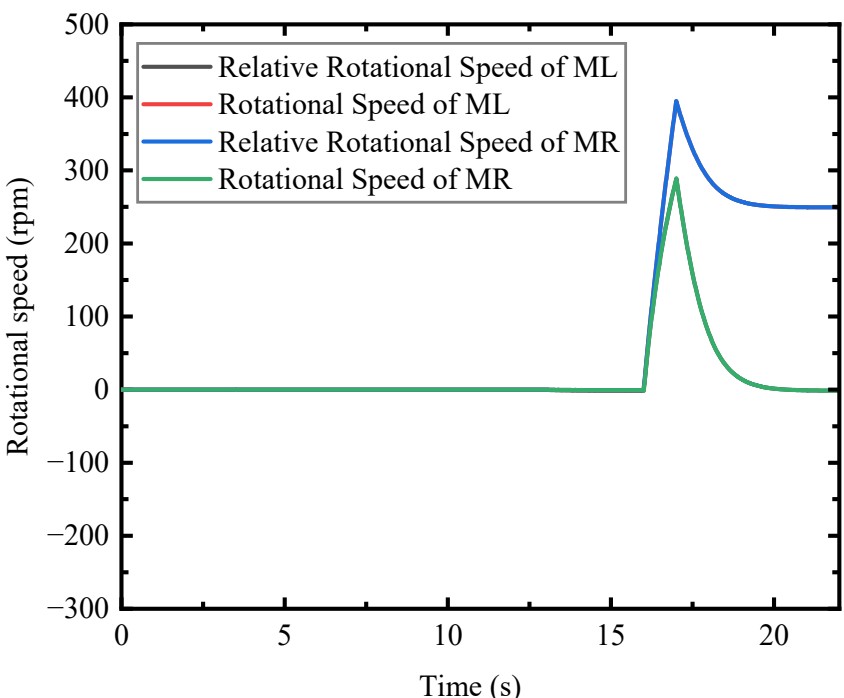

**Figure 19.** Rotational speed of the EDWs and suspension height of the maglev car during static suspension processes.

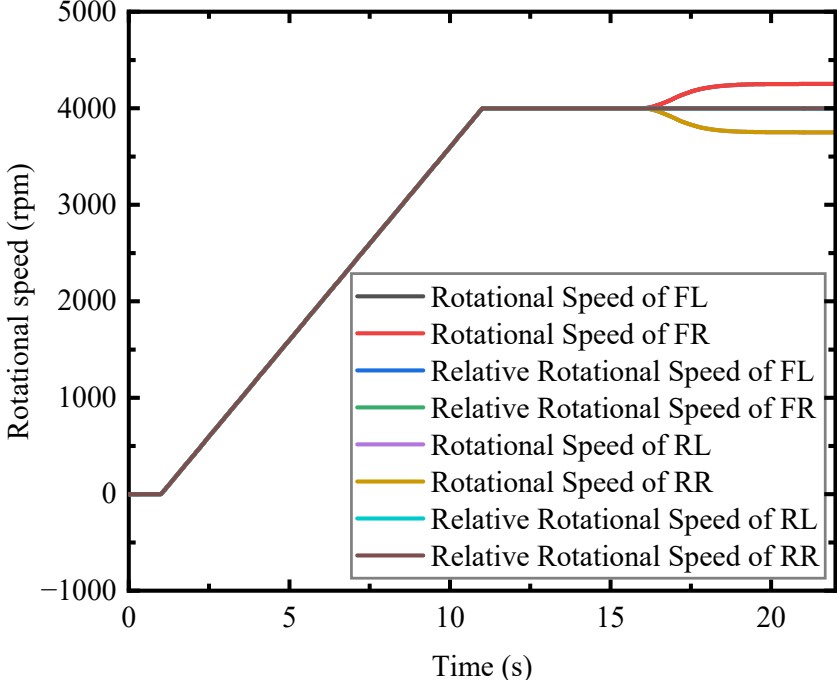

**Figure 20.** Relative rotational speed and rotational speed of the front and rear EDWs during acceleration and constant speed process.

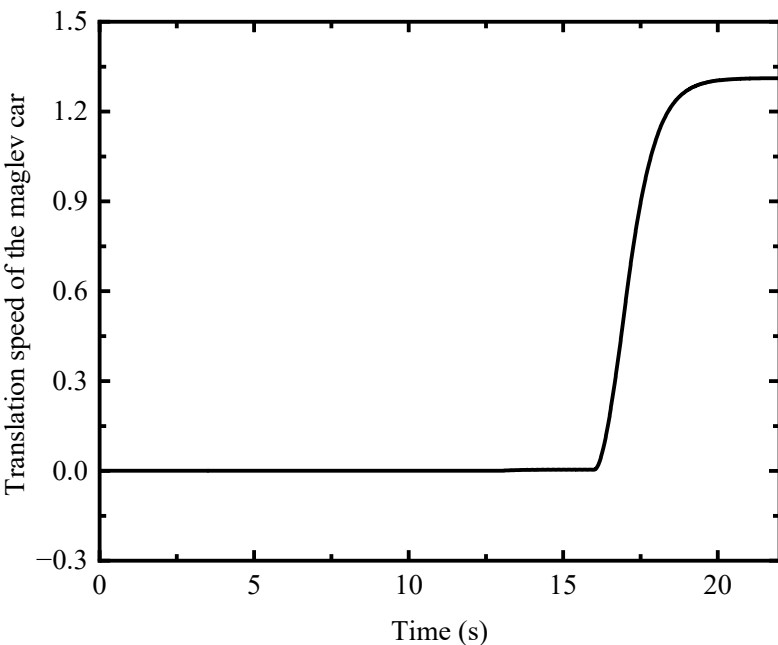

**Figure 21.** Translational speed of the maglev car during acceleration and constant speed process.

### 7.3. Deceleration and Braking

The method of deceleration is as same as the method of acceleration. The speed of the front and rear wheels relative to the ground is kept at 4000 rpm, so that the driving force of the front EDW is equal to the driving force of the rear EDW. Decreasing the rotational speed of the middle EDWs generates the driving force and drives the maglev vehicle to decelerate. The relative rotational speed and rotational speed of the middle EDWs during deceleration and braking processes are shown in Figure 22. The relative rotational speed and rotational speed of the front and rear EDWs during deceleration and braking processes and translation speed of the maglev car during deceleration and braking processes are shown in Figures 23 and 24 respectively.

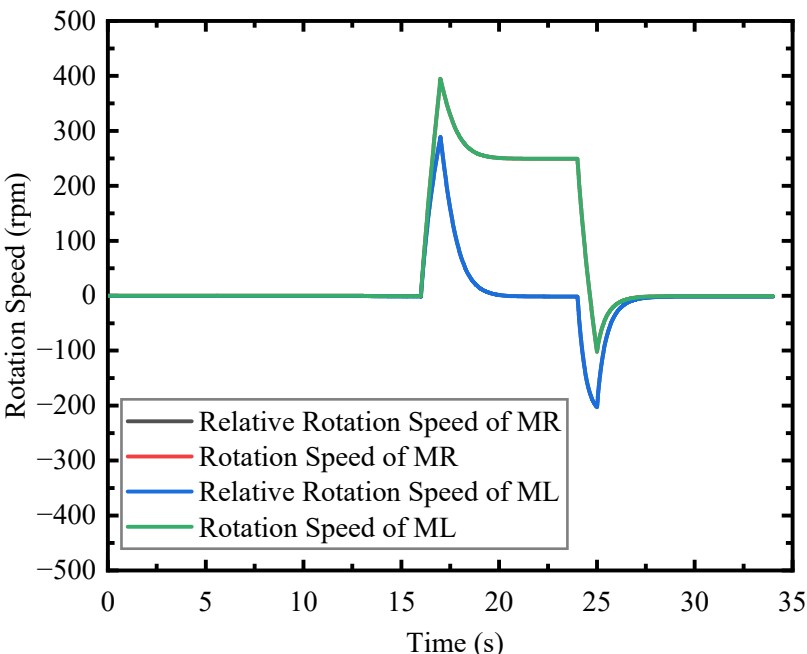

**Figure 22.** Relative rotational speed and rotational speed of the middle EDWs during deceleration and braking processes.

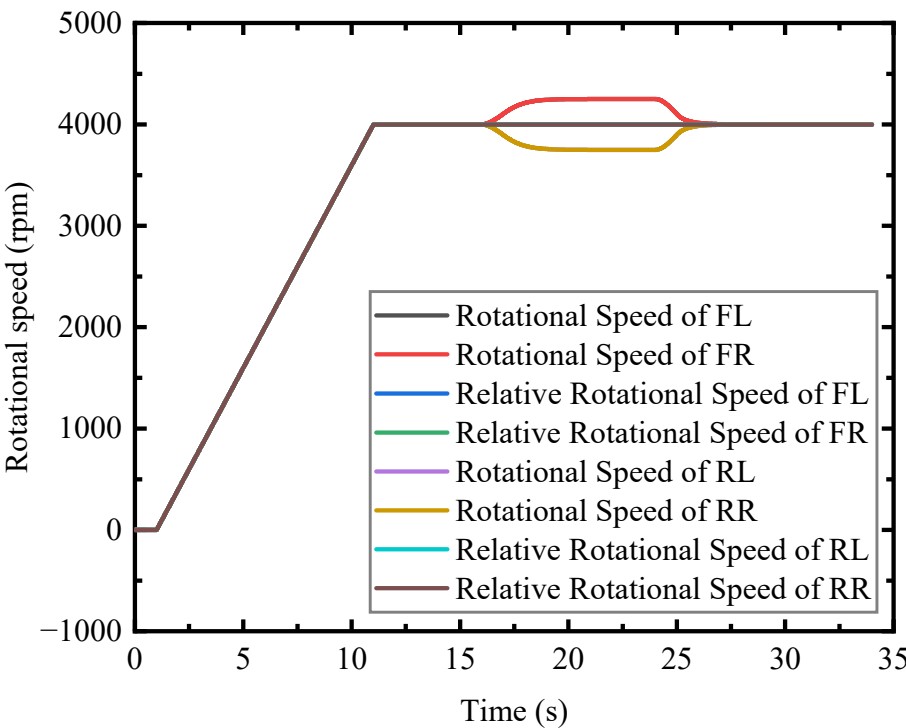

**Figure 23.** Relative rotational speed and rotational speed of the front and rear EDWs during deceleration and braking processes.

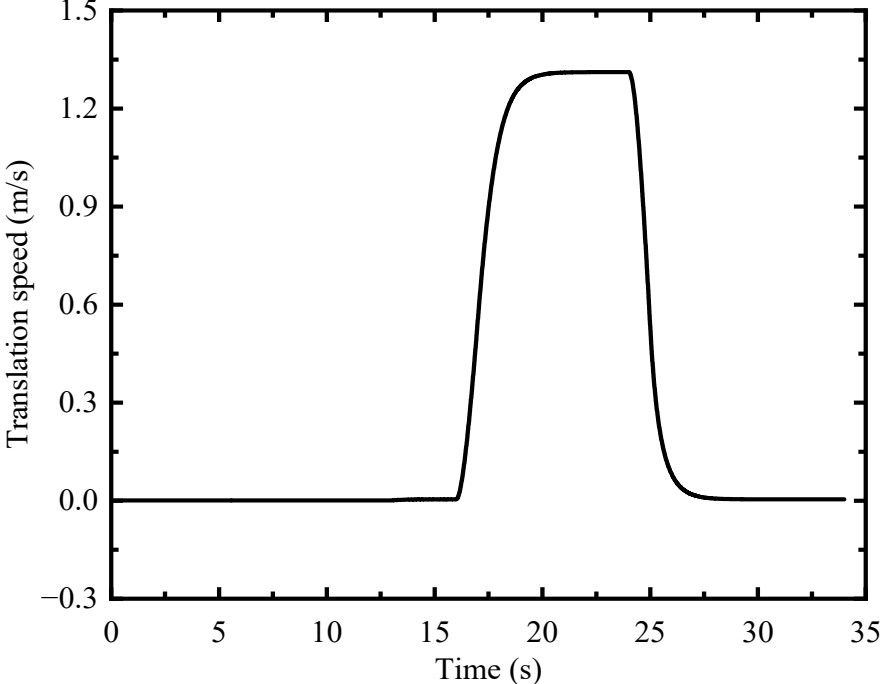

**Figure 24.** Translational speed of the maglev car during deceleration and braking processes.

The first 22 s of the simulation process is the same as the acceleration and uniform speed process. From 24 s to 25 s, reducing the rotational speed of the middle EDWs by 500 rpm. The rotational speed of the middle EDWs reduces from 250 rpm to −102 rpm due to the existence of the vehicle's translation speed and feedback regulation. The relative rotational speed of the middle EDWs became −203 rpm to generate the driving force and drive the maglev vehicle to decelerate. Similarly, the rotational speed of the rear

wheels increases and the rotational speed of the front wheels decreases to keep the relative rotational speed of the front and rear EDWs relative to the induction plate remain 4000 rpm. The driving forces generated by the front and rear EDWs cancel each other. From 25 s to 29.5 s, the translational speed of the maglev vehicle and the relative rotational speed of the middle EDWs gradually decreases to zero and the driving force also becomes zero at 29.5 s. The translational speed of the maglev vehicle becomes 0. The maglev vehicle is at static suspension and realizes the braking function.

### 7.4. Pivot Steering

To achieve the process of pivot steering, it is necessary to keep the relative rotational speed of the front and rear EDWs at 4000 rpm. The rotational speed of the middle EDWs is equal in magnitude and opposite in direction, which will make the maglev car have a rotational speed w. The relative rotational speed and rotational speed of the middle right EDWs are shown in Figure 25 and the rotation angle of the maglev car is shown in Figure 26. This steering method can realize the car's pivot steering by reducing the car's turning radius. In large cities with lots of vehicles and narrow road sections, it can greatly reduce the driver's driving difficulty.

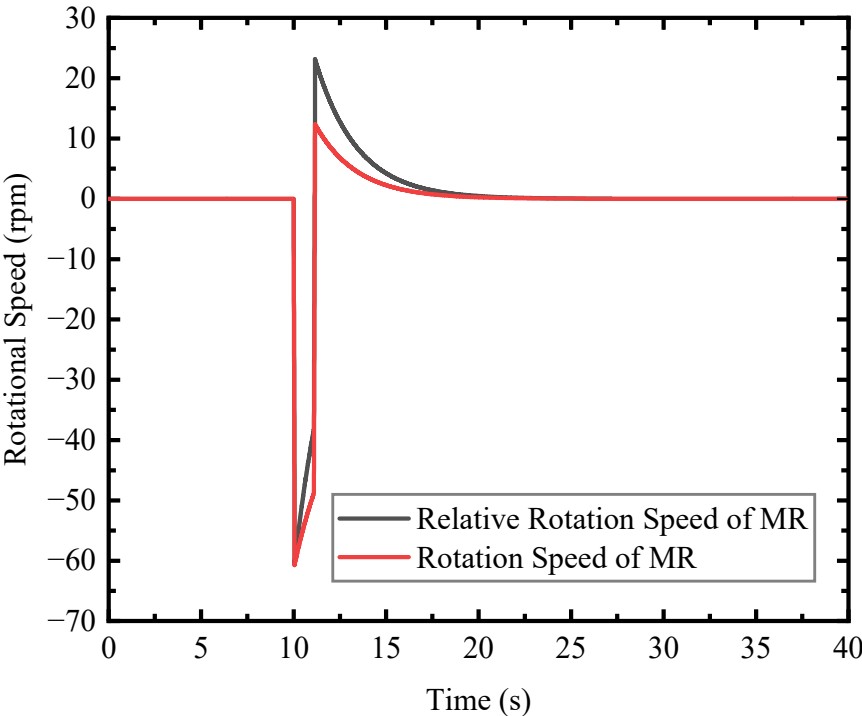

**Figure 25.** Relative rotational speed and rotational speed of the middle right EDWs during pivot steering process.

The first 10 s of the simulation process is as same as the static suspension process. At 10 s the rotational speed of the middle right EDWs decreases from 0 rpm to −61 rpm, and the rotational speed of the middle left EDWs increases from 0 rpm to 61 rpm. The driving forces generated by the right and left EDWs in the middle are equal in magnitude and opposite in direction, so torque is formed, which causes the maglev vehicle to rotate. The rotational speed of the middle EDWs relative to the conductor plate decreases due to the existence of the vehicle's rotational speed and feedback regulation. During this period, the relative rotational speed of the front and rear EDWs remains 4000 rpm, so the driving force of the front EDW is equal to the driving force of the rear EDW and the translation speed of the maglev vehicle gradually remains to zero.

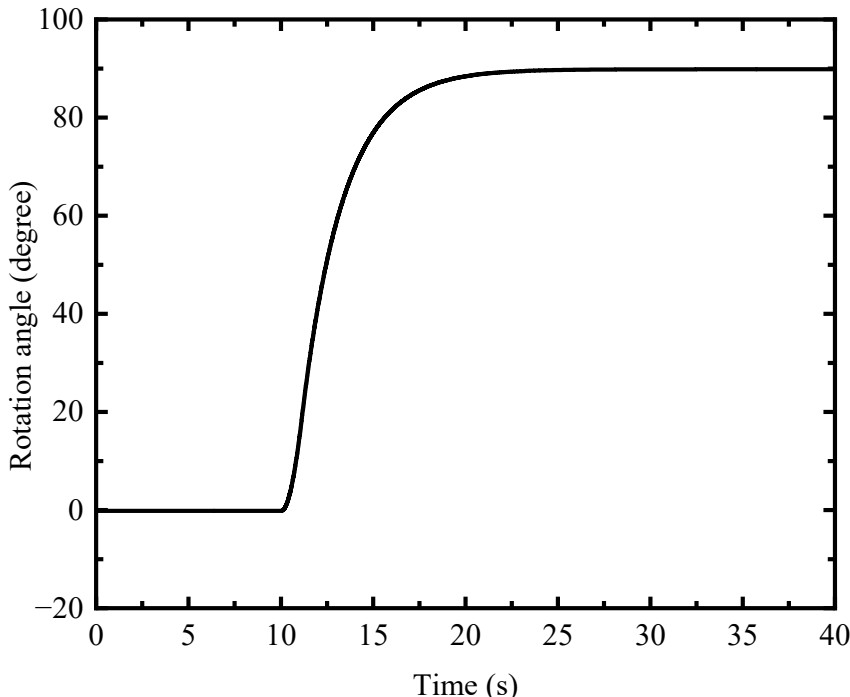

**Figure 26.** Rotation angle of the maglev car during pivot steering process.

At 11.1 s, the rotational speed of the middle right EDWs increases by 61 rpm, and the rotational speed of the middle left EDWs reduces by 61 rpm. The relative rotational speed of the middle right EDWs becomes 23 rpm and the relative rotational speed of the middle lift EDWs becomes $-23$ rpm. The relative rotational speed of the middle EDWs gradually becomes zero from 11.1 s to 27 s. At 27 s, the rotational speed of the middle EDWs relative to the induction plate becomes 0, and the maglev car stops rotating and is at static suspension. The rotational angle of the maglev car is 90 degrees.

The results of the co-simulation of the Simulink and Adams platform shows that the maglev car can successfully achieve the functions of static suspension, acceleration, uniform speed, deceleration, braking, and pivot steering. By increasing the rotational speed of the middle EDWs from 0 to 500 rpm, the maglev vehicle can accelerate and automatically enter the state of uniform speed. The maximum speed is 1.31 m/s. If we change the 1:50 model to a full-scale model, the final speed of the maglev vehicle can be increased. By decreasing the rotational speed of the middle EDWs by 500 rpm, the maglev vehicle can automatically decelerate and enter static suspension state. By changing the speed of the middle EDWs from 0 to 61 rpm, the function of the pivot steering can be realized. the maglev car can turn 90 degrees. All the functions are automatically achieved because of the new feedback control strategy. We only have to change the rotational speed of the middle EDWs. These above integrated operation functions mean this proposed levitation car has the potential to become a means of transportation in the future.

## 8. Conclusions

This paper presents a conceptual model of the permanent magnet electrodynamic suspension vehicle with six wheels based on the annular Halbach. A 1:50 dynamic model of the maglev vehicle is built with co-simulation of Simulink and Adams to verify the feasibility of a new feedback control strategy and study the dynamic characteristics of the maglev vehicle. Our detailed conclusion is summarized by the following points.

1. The maglev car can maintain a suspension air gap at about 14 mm when the rotational speed of the EDWs is 4000 rpm;
2. By changing the rotational speed of the middle EDWs, the maglev vehicle can automatically enter the state of uniform speed and achieve braking. The maximum speed

is 1.31 m/s (1:50 model). By using a full-scale model, the final speed of the maglev vehicle can be increased. The results show that the maglev vehicle can run smoothly using the new feedback control strategy;

3. By changing the speed of the middle EDWs from 0 to 61 rpm, the maglev car can turn 90 degrees. The function of pivot steering can be achieved, which can reduce the car's turning radius and the driver's driving difficulty.

The feasibility of a maglev vehicle with six EDWs has been verified using a co-simulation. Our research can provide a valuable reference for the future design of a new maglev transportation system.

**Author Contributions:** Conceptualization and Supervision Z.D.; software, P.L.; formal analysis, P.L., Z.K., X.W., K.R. and Z.D.; investigation, P.L and W.L. All authors have read and agreed to the published version of the manuscript.

**Funding:** This work was funded by the Sichuan Science and Technology Program (22CXTD0070) and the Science and Technology Program of the Jiangsu Provincial Department of Transport (2021Y02).

**Data Availability Statement:** Not applicable.

**Conflicts of Interest:** The authors declare no conflict of interest.

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
