# Peer review of "Dynamic Characteristics and Working Modes of Permanent Magnet Electrodynamic Suspension Vehicle System Based on Six Wheels of Annular Halbach Structure"

_technologies, doi:10.3390/technologies11010016_

Round 1

Reviewer 1 Report

This is an interesting and well-written paper. Its structure is correct. I formulated some remarks for selected elements.

Enhancement of the references is needed, especially considering the journal “Technologies” as well as other journals from the MDPI group.

New and actual references would be found using specific keywords, especially the keywords used in the manuscript.

For example:

“Maglev” results with 122 papers from the whole MDPI group (one of them is already cited) and 1 in “Technologies” (already cited);

“Halbach Structure” results with 28 papers from the whole MDPI group and 0 in “Technologies”;

“Feedback Control Strategy” results with 542 papers from the whole MDPI group and 1 in “Technologies”.

Some of the sources identified above could be cited in the paper.

The low number of similar publications in the journal “Technologies” could be the result of the selection of not appropriate keywords or the smaller fit to the topics of this journal. Enlargement of the sources and checking the keywords should eliminate this problem.

Finally some smaller remarks.

Some figures are almost identic (3 and 17, 4 and 6, 5 and 26). Please, minimize the number of figures.

I miss more commentary (in conclusions) about the usefulness of the proposed car. Will this solution appropriate vehicle for the future in the context of sustainable transportation? What with the costs of creating the infrastructure for such kind of vehicle?

Will be more sensible to consider bigger vehicles (rather like mass transit of shared-form)? Please, add a commentary.

Author Response

Dear reviewer,

Thank you for your critique. Your comments and suggestions have been helpful in improving the quality of this manuscript. Based on your feedback, we have added some content along with 11 new references and addressed some important details regarding the figures and the explanations in the text. The modifications of the text are highlighted in red in the revised manuscript. The reviewer’s comments are addressed individually hereinafter.

  1. Enhancement of the references is needed, especially considering the journal “Technologies” as well as other journals from the MDPI group. New and actual references would be found using specific keywords, especially the keywords used in the manuscript.

Response: Thank you for your critique. We didn’t cite enough references, especially for the references of the MDPI group and “Technologies”. We have added 11 new references. These are the references that we just add.

[4] Field Measurements and Analyses of Traction Motor Noise of Medium and Low Speed Maglev Train

[17] Performance Simulation of Long-Stator Linear Synchronous Motor for High-Speed Maglev Train under Three-Phase Short-Circuit Fault

[21] Characteristics Analysis of an Electromagnetic Actuator for Magnetic Levitation Transportation

[22] Design and Feasibility Study of MRG-Based Variable Stiffness Soft Robot.

[23] Optimal Design of a New Rotating Magnetic Beacon Structure Based on Halbach Array

[24] Multi-Objective Optimization Design of a Stator Coreless Multidisc Axial Flux Permanent Magnet Motor

[25] Design and Analysis of a Novel Axial-Radial Flux Permanent Magnet Machine with Halbach-Array Permanent Magnets

[28] Simulation of the Braking Effects of Permanent Magnet Eddy Current Brake and Its Effects on Levitation Characteristics of HTS Maglev Vehicles

[30] Characteristic Analysis of the Peak Braking Force and the Critical Speed of Eddy Current Braking in a High-Speed Maglev

[36] Magnetic Circuit Analysis of Halbach Array and Improvement of Permanent Magnetic Adsorption Device for Wall-Climbing Robot

[38] Characteristic Analysis, Simulation, and Experimental Comparison of Two Kinds of Circular Magnet Array in Energy Harvesting

  1. Some figures are almost identic (3 and 17, 4 and 6, 5 and 26). Please, minimize the number of figures.

Response: Thank you for the nice comment. I have deleted Figure 6 and Figure 26. But I didn’t delete Figure 17. I use these two Figure to explain different problems. If I delete one of the Figure 3 or 17, I think that will difficult for the readers to understand.

  1. Will this solution appropriate vehicle for the future in the context of sustainable transportation?

Response: Thank you for your advice. Some content was added to explain this problem.

The six wheels of the car will be divided into suspended wheels, and guided wheels. The EDWs in the middle of the vehicle will provide the driving force for acceleration, deceleration and steering. The EDWs of the front and rear will only provide levitation force for the vehicle. This structure takes advantage of the fact that the magnetic resistance is larger at low speeds and smaller at high speeds. The high-speed magnetic wheels are used to pro-vide levitation and reduce the energy loss caused by magnetic resistance. The less energy loss meets the standards of sustainable transportation. The low-speed magnetic wheels providing the driving force convert the inherent magnetic resistance into driving force, so as to realize the non-contact running of the suspension system. Moreover, the driving energy of the car comes from the rotating motor, which does not use fossil energy. It also meets the requirements of environmental protection, so the maglev vehicle is suitable in the context of sustainable transportation.

  1. What with the costs of creating the infrastructure for such kind of vehicle?

Response: Thank you for your suggestion. Some content was added to explain this problem.

At the same time, the non-magnetic passive track structure can be more easily integrated into the existing traffic infrastructure. The construction cost will not increase too much. In addition, a novel control strategy is proposed in this paper to realize straight-line motion and pivot steering.

  1. Will be more sensible to consider bigger vehicles (rather like mass transit of shared-form)? 

Response: That’s a good question. This needs to be analyzed according to the traffic conditions of cities. In my opinion, large cities are more suitable for mass transit of shared-form, such as Beijing; Smaller cities are more suitable for bigger vehicles.

Reviewer 2 Report

This paper proposes a new maglev vehicle utilizing six EDWs to provide driving force and levitation force. The paper is well-written and organized. Comments to the authors:

1) Comment on the regenerative braking system and how it is applicable to the present system.

2) Highlight the importance of six wheels and its impact on the total cost and safety.

3) The impact of Levitation and thrust Forces on the design can be explained in a better way.

4) Give a suitable reference for the parameters of the permanent magnet EDW and how they are selected. Explain their impact.

5) Mention the software used for the data acquisition system?

6) Highlight the impact of the air gap on the system performance.

7) Add a separate paragraph about the discussion on obtained results.

Author Response

Dear reviewer,

Thank you for your critique. Your comments and suggestions have been helpful in improving the quality of this manuscript. Based on your feedback, we have added some content along with 11 new references and addressed some important details regarding the figures and the explanations in the text. The modifications of the text are highlighted in red in the revised manuscript. The reviewer’s comments are addressed individually hereinafter.

  1. Comment on the regenerative braking system and how it is applicable to the present system.

Response: That’s a good question. But I think there might be some misunderstanding. Maybe my description made you misunderstand, but I didn't mention regenerative braking in my article.

  1. Highlight the importance of six wheels and its impact on the total cost and safety.

Response: Thank you for your suggestion. Some content was added to explain this problem.

However, the traditional four-wheel vehicle structure has a shortcoming. The traditional four-wheel vehicle is difficult to realize the steering function. Because the traditional four-wheel structure changes the magnetic resistance by changing the rotational speed to form a torque, but this method will cause the suspension force of the four EDWs to be inconsistent, resulting in the change of the suspension air gap, which will affect the smoothness of the vehicle operation and lead to steering failure. To solve the steering problem of maglev vehicles, in this paper, we design a new concept maglev vehicle structure based on the radial Halbach permanent magnet array. Figure 1 shows its concept diagram. Although the use of six wheels will increase the overall cost of the vehicle, considering that this structure can help the car achieve steering, these increased costs are worthwhile, and the increase of guided wheels will not have any impact on the driving safety of the car.

  1. The impact of Levitation and thrust Forces on the design can be explained in a better way.

Response: Thank you for your critique. Some content was added to explain this problem and some content has been changed.

Line 107-108: If the rotation speed decreases, the levitation force will decrease and air gap also need to decrease to maintain enough levitation force.

Line 114-125: First, when the maglev vehicle is accelerated by adjusting the rotational speed, it is necessary for traditional four-wheel vehicle structure to decrease the rotational speed of the front EDWs in order to get enough driving force. But this will decrease the levitation force and the suspension gap of the front wheels. If the suspension gap decreases, the driving safety of the vehicle can’t be ensured. Second, the relative rotational speed of the front and rear wheels is not equal when the maglev vehicle is accelerated and decelerated, therefore the suspension force generated by the front and rear wheels are different and a torque is formed, which makes the maglev vehicle has a trend of pitch movement and difficult to run smoothly. Third, although a torque can be formed by decreasing the rotational speed of the EDWs on the diagonal, the suspension air gaps of the four EDWs are different, which will affect the smoothness of the vehicle operation and lead to steering failure.

  1. Give a suitable reference for the parameters of the permanent magnet EDW and how they are selected. Explain their impact.

Response: Thank you for your critique. Some content was added to explain this problem

The levitation force of a circular Halbach array is affected by many factors. The size, material, magnetization angle of the magnet, and the number of pole pairs are factors that affect the levitation force of the Halbach array. However, in order to improve the suspension height, we need to improve the suspension force without changing the wheel weight. Increasing the size of magnets can improve the levitation force, but it will increase the weight of the wheel and increase the cost; Selecting better magnet materials can also increase the levitation force, but this kind of magnet is expensive and difficult to produce.

  1. Mention the software used for the data acquisition system?

Response: Thank you for your suggestion. Some content was added to explain it.

Line 191-193: The data acquisition system uses the software designed by LabVIEW. It can display, collect and export force data and distance data in three directions.

  1. Highlight the impact of the air gap on the system performance.

Response: Thank you for your critique. Some content was added to explain this problem.

When the road is irregularity, the vehicle body will vibrate vertically. If the air gap is too small, the EDWs of maglev vehicle will collide with the conductor plate. In order to ensure the driving safety of the vehicle, the air gap of the maglev vehicle must be increased.

  1. Add a separate paragraph about the discussion on obtained results.

Response: Thank you for your advice. Some content was added to explain this problem.

The results of the co-simulation of the Simulink and Adams platform shows that the maglev car can successfully achieve the function of static suspension, acceleration, uniform speed, deceleration, braking, and pivot steering. By increasing the rotational speed of the middle EDWs from 0 to 500 rpm, the maglev vehicle can accelerate and automatically enter the state of the uniform speed. The maximum speed is 1.31 m/s. If we change the 1:50 model to a full-scale model, the final speed of the maglev vehicle can be increased. By decreasing the rotational speed of the middle EDWs by 500 rpm, the maglev vehicle can automatically decelerate and enter static suspension state. By changing the speed of the middle EDWs from 0 to 61 rpm, the function of the pivot steering can be realized. the maglev car can turn 90 degrees. All the functions are automatically achieved because of the new feedback control strategy. We only have to change the rotational speed of the middle EDWs. And these above integrated operation functions make this proposed levitation car have the potential to become a means of transportation in the future.

Reviewer 3 Report

Overall the paper is well written and of interest. The main question addressed by the research is to a new type of a conceptual model of the six-wheel permanent magnet electrodynamic suspension vehicle based on the annular Halbach. A 1:50 dynamic model of the maglev vehicle is built with the co-simulation of Simulink and Adams to verify the feasibility of a new feedback control strategy and to study the dynamic characteristics of the maglev vehicle. Where the originality or relevance in the field and brings it to the field compared to other publications is not very clear. However, we note the absence of validated mathematical formulas describes the observed all physical phenomena and it should be justified. No experimental data or predictions from other calculations are available for comparison. The accuracy and validity the proposed model are therefore unclear. In consequence, the author needs to address the evidence before this reviewer agrees with publication of this paper. Also, a more careful literature review work is suggested. Hence the originality and novelty of manuscript (or the proposed the methods) are not clear.

This is a good paper, but you need to conduct a “Minor revision”. After those corrections the manuscript may be published in the Journal. The following comments are split into some general ones and some more specific comment.

Author Response

Dear reviewer,

Thank you for your critique. Your comments and suggestions have been helpful in improving the quality of this manuscript. Based on your feedback, we have added some content along with 11 new references and addressed some important details regarding the figures and the explanations in the text. The modifications of the text are highlighted in red in the revised manuscript. The reviewer’s comments are addressed individually hereinafter.

  1. We note the absence of validated mathematical formulas describes the observed all physical phenomena and it should be justified. No experimental data or predictions from other calculations are available for comparison. The accuracy and validity the proposed model are therefore unclear.

Response: Thank you for your critique. First, for the problem of validated mathematical formulas. This paper studies six EDWs maglev car, which is a new concept. Most of the previous research is based on the traditional four EDWs maglev car, so the mathematical formulas are still lacking. This paper is mainly an exploratory study, and the mathematical formula needs to be further studied later. Second, for the problem of the experimental data or predictions from other calculations, it is the same problem. It is a new concept, the six EDWs vehicle is still in the initial research stage, so there is no corresponding experimental data or predictions from other calculations. But we will continue to study. If conditions permit, we will also make a test prototype for experimental verification.

  1. For the part of introduction,

(1). The literature cited is relevant to the study

(2). Very limited literature review has been performed. I suggest extending it.

(3). The objective to be achieved by your study is not mentioned here in this section.

Response: Thank you for your critique. We have added 11 references. Here is the list:

[4] Field Measurements and Analyses of Traction Motor Noise of Medium and Low Speed Maglev Train

[17] Performance Simulation of Long-Stator Linear Synchronous Motor for High-Speed Maglev Train under Three-Phase Short-Circuit Fault

[21] Characteristics Analysis of an Electromagnetic Actuator for Magnetic Levitation Transportation

[22] Design and Feasibility Study of MRG-Based Variable Stiffness Soft Robot.

[23] Optimal Design of a New Rotating Magnetic Beacon Structure Based on Halbach Array

[24] Multi-Objective Optimization Design of a Stator Coreless Multidisc Axial Flux Permanent Magnet Motor

[25] Design and Analysis of a Novel Axial-Radial Flux Permanent Magnet Machine with Halbach-Array Permanent Magnets

[28] Simulation of the Braking Effects of Permanent Magnet Eddy Current Brake and Its Effects on Levitation Characteristics of HTS Maglev Vehicles

[30] Characteristic Analysis of the Peak Braking Force and the Critical Speed of Eddy Current Braking in a High-Speed Maglev

[36] Magnetic Circuit Analysis of Halbach Array and Improvement of Permanent Magnetic Adsorption Device for Wall-Climbing Robot

[38] Characteristic Analysis, Simulation, and Experimental Comparison of Two Kinds of Circular Magnet Array in Energy Harvesting

But most of the articles focus on the research of single EDW, there aren’t lots of article about the system of maglev vehicle. We have cited the reference of article that focus on the research of the maglev vehicle system.

For the objective of the article, some content was added to explain this problem.

However, the traditional four-wheel vehicle structure has a shortcoming. The traditional four-wheel vehicle is difficult to realize the steering function. Because the traditional four-wheel structure changes the magnetic resistance by changing the rotational speed to form a torque, but this method will cause the suspension force of the four EDWs to be inconsistent, resulting in the change of the suspension air gap, which will affect the smoothness of the vehicle operation and lead to steering failure. To solve the steering problem of maglev vehicles, in this paper, we design a new concept maglev vehicle structure based on the radial Halbach permanent magnet array.

  1. For the part of conclusion

(1). This conclusion would be much stronger were such a discussion provided.

(2). It would be much better if the authors could summarize main features of the proposed method in the conclusion

(3). Conclusions should be written in more detail adding numeric data.

Response: Thank you for your critique. We have changed the content of the conclusion.

1.The maglev car can remain a suspension air gap at about 14mm when the rotational speed of the EDWs is 4000rpm;

2.By changing the rotational speed of the middle EDWs, the maglev vehicle can automatically enter the state of the uniform speed and realize the braking. The maximum speed is 1.31 m/s (1:50 model). By using a full-scale model, the final speed of the maglev vehicle can be increased. The results show that the maglev vehicle can run smoothly using the new feedback control strategy;

3.By changing the speed of the middle EDWs from 0 to 61 rpm, the maglev car can turn 90 degrees. The function of pivot steering can be achieved, which can reduce the car's turning radius and driver's driving difficulty.

  1. Figure 16 is not clear.

Response: Thank you for your critique. We have changed figure. The number of the figure has changed to 15.
